# MY BODY IS A CAGE: THE ROLE OF MORPHOLOGY IN GRAPH-BASED INCOMPATIBLE CONTROL

**Vitaly Kurin**
Department of Computer Science
University of Oxford
Oxford, United Kingdom
vitaly.kurin@cs.ox.ac.uk

**Maximilian Igl**
Department of Computer Science
University of Oxford
Oxford, United Kingdom
maximilian.igl@eng.ox.ac.uk

**Tim Rocktäschel**
Department of Computer Science
University College London
London, United Kingdom
t.rocktaschel@cs.ucl.ac.uk

**Wendelin Böhmer**
Department of Software Technology
Delft University of Technology
Delft, Netherlands
j.w.bohmer@tudelft.nl

**Shimon Whiteson**
Department of Computer Science
University of Oxford
Oxford, United Kingdom
shimon.whiteson@cs.ox.ac.uk

## ABSTRACT

Multitask Reinforcement Learning is a promising way to obtain models with better performance, generalisation, data efficiency, and robustness. Most existing work is limited to *compatible* settings, where the state and action space dimensions are the same across tasks. Graph Neural Networks (GNN) are one way to address incompatible environments, because they can process graphs of arbitrary size. They also allow practitioners to inject biases encoded in the structure of the input graph. Existing work in graph-based continuous control uses the physical morphology of the agent to construct the input graph, i.e., encoding limb features as node labels and using edges to connect the nodes if their corresponded limbs are physically connected. In this work, we present a series of ablations on existing methods that show that morphological information encoded in the graph does not improve their performance. Motivated by the hypothesis that any benefits GNNs extract from the graph structure are outweighed by difficulties they create for message passing, we also propose AMORPHEUS, a transformer-based approach. Further results show that, while AMORPHEUS ignores the morphological information that GNNs encode, it nonetheless substantially outperforms GNN-based methods that use the morphological information to define the message-passing scheme.

## 1 INTRODUCTION

Multitask Reinforcement Learning (MTRL) (Vithayathil Varghese & Mahmoud, 2020) leverages commonalities between multiple tasks to obtain policies with better returns, generalisation, data efficiency, or robustness. Most MTRL work assumes *compatible* state-action spaces, where the dimensionality of the states and actions is the same across tasks. However, many practically important domains, such as robotics, combinatorial optimization, and object-oriented environments, have *incompatible* state-action spaces and cannot be solved by common MTRL approaches.

Incompatible environments are avoided largely because they are inconvenient for function approximation: conventional architectures expect fixed-size inputs and outputs. One way to overcome this limitation is to use Graph Neural Networks (GNNs) (Gori et al., 2005; Scarselli et al., 2005; Battaglia et al., 2018). A key feature of GNNs is that they can process graphs of arbitrary size and thus, in

principle, allow MTRL in incompatible environments. However, GNNs also have a second key feature: they allow models to condition on structural information about how state features are related, e.g., how a robot's limbs are connected. In effect, this enables practitioners to incorporate additional domain knowledge where states are described as labelled graphs. Here, a graph is a collection of labelled nodes, indicating the features of corresponding objects, and edges, indicating the relations between them. In many cases, e.g., with the robot mentioned above, such domain knowledge is readily available. This results in a structural inductive bias that restricts the model's computation graph, determining how errors backpropagate through the network.

GNNs have been applied to MTRL in continuous control environments, a staple benchmark of modern Reinforcement Learning (RL), by leveraging both of the key features mentioned above (Wang et al., 2018; Huang et al., 2020). In these two works, the labelled graphs are based on the agent's physical morphology, with nodes labelled with the observable features of their corresponding limbs, e.g., coordinates, angular velocities and limb type. If two limbs are physically connected, there is an edge between their corresponding nodes. However, the assumption that it is beneficial to restrict the model's computation graph in this way has to our knowledge not been validated.

To investigate this issue, we conduct a series of ablations on existing GNN-based continuous control methods. The results show that removing morphological information does not harm the performance of these models. In addition, we propose AMORPHEUS, a new continuous control MTRL method based on transformers (Vaswani et al., 2017) instead of GNNs that use morphological information to define the message-passing scheme. AMORPHEUS is motivated by the hypothesis that any benefit GNNs can extract from the morphological domain knowledge encoded in the graph is outweighed by the difficulty that the graph creates for message passing. In a sparsely connected graph, crucial state information must be communicated across multiple hops, which we hypothesise is difficult in practice to learn. AMORPHEUS uses transformers instead, which can be thought of as fully connected GNNs with attentional aggregation (Battaglia et al., 2018). Hence, AMORPHEUS ignores the morphological domain knowledge but in exchange obviates the need to learn multi-hop communication. Similarly, in Natural Language Processing, transformers were shown to perform better without an explicit structural bias and even learn such structures from data (Vig & Belinkov, 2019; Goldberg, 2019; Tenney et al., 2019; Peters et al., 2018).

Our results on incompatible MTRL continious control benchmarks (Huang et al., 2020; Wang et al., 2018) strongly support our hypothesis: AMORPHEUS substantially outperforms GNN-based alternatives with fixed message-passing schemes in terms of sample efficiency and final performance. In addition, AMORPHEUS exhibits nontrivial behaviour such as cyclic attention patterns coordinated with gaits.

## 2 BACKGROUND

We now describe the necessary background for the rest of the paper.

### 2.1 REINFORCEMENT LEARNING

A Markov Decision Process (MDP) is a tuple $\langle \mathcal{S}, \mathcal{A}, \mathcal{R}, \mathcal{T}, \rho_0 \rangle$. The first two elements define the set of states $\mathcal{S}$ and the set of actions $\mathcal{A}$. The next element defines the reward function $\mathcal{R}(s, a, s')$ with $s, s' \in \mathcal{S}$ and $a \in \mathcal{A}$. $\mathcal{T}(s'|s, a)$ is the probability distribution function over states $s' \in \mathcal{S}$ after taking action $a$ in state $s$. The last element of the tuple $\rho_0$ is the distribution over initial states. Task and environment are synonyms for MDPs in this work.

A policy $\pi(a|s)$ is a mapping from states to distributions over actions. The goal of an RL agent is to find a policy that maximises the expected discounted cumulative return $J = \mathbb{E}\big[\sum_{t=0}^{\infty} \gamma^t r_t\big]$, where $\gamma \in [0, 1)$ is a discount factor, $t$ is the discrete environment step and $r_t$ is the reward at step $t$. In the MTRL setting, the agent aims to maximise the average performance across $N$ tasks: $\frac{1}{N}\sum_{i=1}^{N} J_i$. We use MTRL *return* to denote the average performance across the tasks.

In this paper, we assume that states and actions are multivariate, but dimensionality remains constant for one MDP: $s \in \mathbb{R}^k, \forall s \in \mathcal{S}$, and $a \in \mathbb{R}^{k'}, \forall a \in \mathcal{A}$. We use $dim(\mathcal{S}) = k$ and $dim(\mathcal{A}) = k'$ to denote this dimensionality, which can differ amongst MDPs. We consider two tasks $MDP_1$ and $MDP_2$ as *incompatible* if the dimensionality of their state or action spaces disagree, i.e., $dim(\mathcal{S}_1) \neq$

$dim(\mathcal{S}_2)$ or $dim(\mathcal{A}_1) \neq dim(\mathcal{A}_2)$ with the subscript denoting a task index. In this case MTRL policies or value functions can not be represented by a Multi-layer Perceptron (MLP), which requires fixed input dimensions. We do not have additional assumptions on the semantics behind the state and action set elements and focus on the dimensions mismatch only.

Our approach, as well as the baselines in this work (Wang et al., 2018; Huang et al., 2020), use Policy Gradient (PG) methods (Peters & Schaal, 2006). PG methods optimise a policy using gradient ascent on the objective: $\theta_{t+1} = \theta_t + \alpha \nabla_\theta J|_{\theta=\theta_t}$, where $\theta$ parameterises a policy. Often, to reduce variance in the gradient estimates, one learns a critic so that the policy gradient becomes $\nabla_\theta J(\theta) = \mathbb{E}\left[\sum_t A_t^\pi \nabla_\theta \log \pi_\theta(a_t|s_t)\right]$, where $A_t^\pi$ is an estimate of the advantage function (e.g., TD residual $r_t + \gamma V^\pi(s_{t+1}) - V^\pi(s_t)$). The state-value function $V^\pi(s)$ is the expected discounted return a policy $\pi$ receives starting at state $s$. Wang et al. (2018) use PPO (Schulman et al., 2017), which restricts a policy update to avoid instabilities from drastic changes in the policy behaviour. Huang et al. (2020) use TD3 (Fujimoto et al., 2018), a PG method based on DDPG (Lillicrap et al., 2016).

## 2.2 GRAPH NEURAL NETWORKS FOR INCOMPATIBLE MULTITASK RL

GNNs can address incompatible environments because they can process graphs of arbitrary sizes and topologies. A GNN is a function that takes a labelled graph as input and outputs a graph $\mathcal{G}'$ with different labels but the same topology. Here, a labelled graph $\mathcal{G} := \langle \mathcal{V}, \mathcal{E} \rangle$ consists of a set of vertices $v^i \in \mathcal{V}$, labelled with vectors $\boldsymbol{v}^i \in \mathbb{R}^{m_v}$ and a set of directed edges $e^{ij} \in \mathcal{E}$ from vertex $v^i$ to $v^j$, labelled with vectors $\boldsymbol{e}^{ij} \in \mathbb{R}^{m_e}$. The output graph $\mathcal{G}'$ has the same topology but the labels can be of different dimensionality than the input, that is, $\boldsymbol{v}'^i \in \mathbb{R}^{m'_v}$ and $\boldsymbol{e}'^{ij} \in \mathbb{R}^{m'_e}$. By graph topology we mean the connectivity of the graph, which can be represented by an adjacency matrix, a binary matrix $\{a\}_{ij}$ whose elements $a_{ij}$ equal to one iff there is an edge $e_{ij} \in \mathcal{E}$ connecting vertices $v_i, v_j \in \mathcal{V}$.

A GNN computes the output labels for entities of type $k$ by parameterised *update functions* $\phi_\psi^k$ represented by neural networks that can be learnt end-to-end via backpropagation. These updates can depend on a varying number of edges or vertices, which have to be summarised first using *aggregation functions* that we denote $\rho$. Apart from their ability to operate on sets of elements, aggregation functions should be permutation invariant. Examples of such aggregation functions include summation, averaging and $\max$ or $\min$ operations.

Incompatible MTRL for continuous control implies learning a common policy for a set of agents with different number of limbs and connectivity of those limbs, i.e. *morphology*. To be more precise, a set of incompatible continuous control environments is a set of MDPs described in Section 2.1. When a state is represented as a graph, each node label contains features of its corresponding limb, e.g., limb type, coordinates, and angular velocity. Similarly, each factor of an action set element corresponds to a node with the label meaning the torque for a joint to emit. The typical reward function of a MuJoCo (Todorov et al., 2012) environment includes a reward for staying alive, distance covered, and a penalty for action magnitudes.

We now describe two existing approaches to incompatible control: NERVENET (Wang et al., 2018) and Shared Modular Policies (SMP) (Huang et al., 2020).

### 2.2.1 NERVENET

In NERVENET, the input observations are first encoded via a MLP processing each node labels as a batch element: $\boldsymbol{v}^i \leftarrow \phi_\chi(\boldsymbol{v}^i), \forall v^i \in \mathcal{V}$. After that, the message-passing part of the model block performs the following computations (in order):

$$
\begin{aligned}
\boldsymbol{e}'^{ij} &\leftarrow \phi_\psi^e(\boldsymbol{v}^i) & ,\forall e^{ij} \in \mathcal{E}, \\
\boldsymbol{v}^i &\leftarrow \phi_\xi^v(\boldsymbol{v}^i, \rho\{\boldsymbol{e}'^{ki} \,|\, e^{ki} \in \mathcal{E}\}) & ,\forall v^i \in \mathcal{V}.
\end{aligned}
$$

The edge updater $\phi_\psi^e$ in NERVENET is an MLP which does not take the receiver's state into account. Using only one message pass restricts the learned function to local computations on the graph. The node updater $\phi_\xi^v$ is a Gated Recurrent Unit (GRU) (Cho et al., 2014) which maintains the internal state when doing multiple message-passing iterations, and takes the aggregated outputs of the edge updater for all incoming edges as inputs. After the message-passing stage, the MLP decoder takes the states of the nodes and, like the encoder, independently processes them, emitting scalars used as

the mean for the normal distribution from which actions are sampled: $\boldsymbol{v}_{dec}^i \leftarrow \phi_\eta(\boldsymbol{v}^i), \forall v^i \in \mathcal{V}$. The standard deviation of this distribution is a separate state-independent vector with one scalar per action.

### 2.2.2 SHARED MODULAR POLICIES

SMP is a variant of a GNN that operates only on trees. Computation is performed in two stages: top-down and bottom-up. In the first stage, information propagates level by level from leaves to the root with parents aggregating information from their children. In the second stage, information propagates from parents to the leaves with parents emitting multiple messages, one per child. The policy emits actions at the second stage of the computation together with the downstream messages.

Instead of a permutation invariant aggregation, the messages are concatenated. This, as well as separate messages for the children, also injects structural bias to the model, e.g., separating the messages for the left and right parts of robots with bilateral symmetry. In addition, its message-passing schema depends on the morphology and the choice of the root node. In fact, Huang et al. (2020) show that the root node choice can affect performance by 15%.

SMP trains a separate model for the actor and critic. An actor outputs one action per non-root node. The critic outputs a scalar per node as well. When updating a critic, a value loss is computed independently per each node with targets using the same scalar reward from the environment.

### 2.3 TRANSFORMERS

Transformers can be seen as GNNs applied to fully connected graphs with the attention as an edge-to-vertex aggregation operation (Battaglia et al., 2018). Self-attention used in transformers is an associative memory-like mechanism that first projects the feature vector of each node $\boldsymbol{v}^i \in \mathbb{R}^{m_v}$ into three vectors: query $\boldsymbol{q}_i := \boldsymbol{\Theta} \boldsymbol{v}^i \in \mathbb{R}^d$, key $\boldsymbol{k}_i := \bar{\boldsymbol{\Theta}} \boldsymbol{v}^i \in \mathbb{R}^d$ and value $\hat{\boldsymbol{v}}_i := \hat{\boldsymbol{\Theta}} \boldsymbol{v}^i \in \mathbb{R}^{m_v}$. Parameter matrices $\boldsymbol{\Theta}, \bar{\boldsymbol{\Theta}}$, and $\hat{\boldsymbol{\Theta}}$ are learnt. The query of the receiver $v_i$ is compared to the key value of senders using a dot product. The resulting values $\boldsymbol{w}_i$ are used as weights in the weighted sum of all the value vectors in the graph. The computation proceeds as follows:

$$
\begin{aligned}
\boldsymbol{w}_i &:= \text{softmax}\left(\frac{[\boldsymbol{k}_1,\ldots,\boldsymbol{k}_n]^\top \boldsymbol{q}_i}{\sqrt{d}}\right) \quad, \forall v_i \in \mathcal{V}\,, \\
\boldsymbol{v}_i' &:= [\hat{\boldsymbol{v}}_1,\ldots,\hat{\boldsymbol{v}}_n]\boldsymbol{w}_i
\end{aligned}
\tag{1}
$$

with $[x_1, x_2, ..., x_n]$ being a $\mathbb{R}^{k \times n}$ matrix of concatenated vectors $x_i \in \mathbb{R}^k$. Often, multiple attention heads, i.e., $\boldsymbol{\Theta}, \bar{\boldsymbol{\Theta}}$, and $\hat{\boldsymbol{\Theta}}$ matrices, are used to learn different interactions between the nodes and mitigate the consequences of unlucky initialisation. The output of multiple heads is concatenated and later projected to respect the dimensions.

A transformer block is a combination of an attention block and a feedforward layer with a possible normalisation between them. In addition, there are residual connections from the input to the attention output and from the output of the attention to the feedforward layer output. Transformer blocks can be stacked together to take higher order dependencies into account, i.e., reacting not only to the features of the nodes, but how the features of the nodes change after applying a transformer block.

## 3 THE ROLE OF MORPHOLOGY IN EXISTING WORK

In this section, we provide evidence against the assumption that GNNs improve performance by exploiting information about physical morphology (Huang et al., 2020; Wang et al., 2018). Here and in all of the following sections, we run experiments for three random seeds and report the average undiscounted MTRL return and the standard error across the seeds.

To determine if information about the agent's morphology encoded in the relational graph structure is essential to the success of SMP, we compare its performance given full information about the structure (morphology), given no information about the structure (star), and given a structural bias unrelated to the agent's morphology (line). Ideally, we would test a fully connected architecture as well, but SMP only works with trees. Figure 9 in Appendix B illustrates the tested topologies.

The results in Figure 1a and 1b demonstrate that, surprisingly, performance is not contingent on having information about the physical morphology. A `star` agent performs on par with the

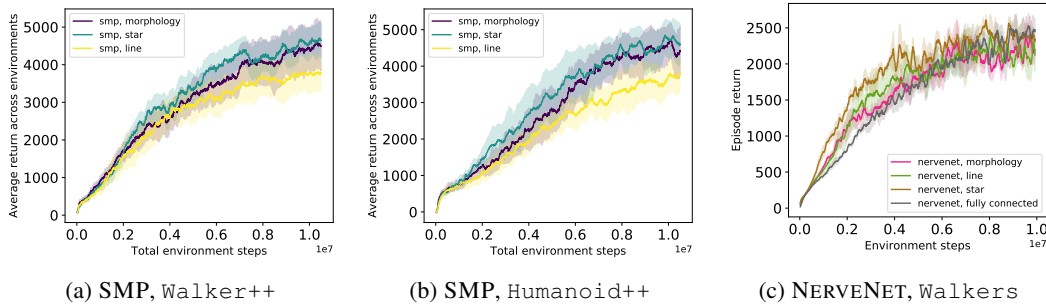

(a) SMP, `Walker++`       (b) SMP, `Humanoid++`       (c) NERVENET, `Walkers`

Figure 1: Neither SMP nor NERVENET leverage the agent's morphological information, or the positive effects are outweighted by their negative effect on message passing.

`morphology` agent, thus refuting the assumption that the method learns because it exploits information about the agent's physical morphology. The `line` agent performs worse, perhaps because the network must propagate messages even further away, and information is lost with each hop due to the finite size of the MLPs causing information bottlenecks (Alon & Yahav, 2020).

We also present similar results for NERVENET. Figure 1c shows that all of the variants we tried perform similarly well on `Walkers` from (Wang et al., 2018), with `star` being marginally better. Since NERVENET can process non-tree graphs, we also tested a fully connected variant. This version learns more slowly at the beginning, probably because of difficulties with differentiating nodes at the aggregation step. Interestingly, in contrast to SMP, in NERVENET `line` performs on par with `morphology`. This might be symptomatic of problems with the message-passing mechanism of SMP, e.g., bottlenecks leading to information loss.

# 4 AMORPHEUS

Inspired by the results above, we propose AMORPHEUS, a transformer-based method for incompatible MTRL in continuous control. AMORPHEUS is motivated by the hypothesis that any benefit GNNs can extract from the morphological domain knowledge encoded in the graph is outweighed by the difficulty that the graph creates for message passing. In a sparse graph, crucial state information must be communicated across multiple hops, which we hypothesise is difficult to learn in practice.

AMORPHEUS belongs to the encode-process-decode family of architectures (Battaglia et al., 2018) with a transformer at its core. Since transformers can be seen as GNNs operating on fully connected graphs, this approach allows us to learn a message passing schema for each state and each pass separately, and limits the number of message passes needed to propagate sufficient information through the graph. Multi-hop message propagation in the presence of aggregation, which could cause problems with gradient propagation and information loss, is no longer required. We implement both actor and critic in the SMP codebase (Huang et al., 2020) and made our implementation available online at `https://github.com/yobibyte/amorpheus`. Like in SMP, there is no weight

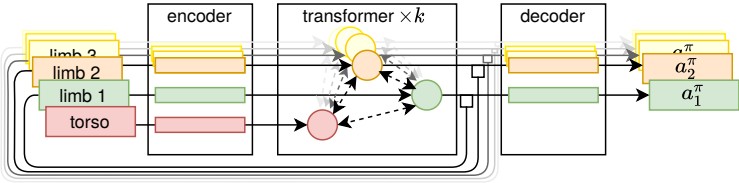

Figure 2: AMORPHEUS architecture. Lines with squares at the end denote concatenation. Arrows going separately through encoder and decoder denote that rows of the input matrix are processed independently as batch elements. Dashed arrows denote message-passing in a transformer block. The diagram depicts the policy network, the critic has an identical architecture, with the decoder outputs interpreted as value function values.

sharing between the actor and the critic. Both of them consist of three parts: a linear encoder, a transformer in the middle, and the output decoder MLP.

Figure 2 illustrates the AMORPHEUS architecture (policy). The encoder and decoder process each node independently, as if they are different elements of a mini-batch. Like SMP, the policy network has one output per graph node. The critic has the same architecture as on Figure 2, and, as in Huang et al. (2020), each critic node outputs a scalar with the value loss independently computed per node.

Similarly to NERVENET and SMP, AMORPHEUS is modular and can be used in incompatible environments, including those not seen in training. In contrast to SMP which is constrained by the maximum number of children per node seen at the model initialisation in training, AMORPHEUS can be applied to any other morphology with no constraints on the physical connectivity.

Instead of one-hot encoding used in natural language processing, we apply a linear layer on node observations. Each node observation uses the same state representation as SMP and includes a limb type (e.g. hip or shoulder), position with a relative $x$ coordinate of the limb with respect to the torso, positional and rotational velocities, rotations, angle and possible range of the values for the angle normalised to $[0, 1]$. We add residual connections from the input features to the decoder output to avoid the nodes forgetting their own features by the time the decoder independently computes the actions. Both actor and critic use two attention heads for each of the three transformer layers. Layer Normalisation (Ba et al., 2016) is a crucial component of transformers which we also use in AMORPHEUS. See Appendix A for more details on the implementation.

## 4.1 EXPERIMENTAL RESULTS

We first test AMORPHEUS on the set of MTRL environments proposed by Huang et al. (2020). For `Walker++`, we omit flipped environments, since Huang et al. (2020) implement flipping on the model level. For AMORPHEUS, the flipped environments look identical to the original ones. Our experiments in this Section are built on top of the TD3 implementation used in Huang et al. (2020).

Figure 3 supports our hypothesis that explicit morphological information encoded in graph topology is not needed to yield a single policy achieving high average returns across a set of incompatible continuous control environments. Free from the need to learn multi-hop communication and equipped with the attention mechanism, AMORPHEUS clearly outperforms SMP, the state-of-the-art algorithm for incompatible continuous control. Huang et al. (2020) report that training SMP on `Cheetah++` together with other environments makes SMP unstable. By contrast, AMORPHEUS has no trouble learning in this regime (Figure 3g and 3h).

Our experiments demonstrate that node features have enough information for AMORPHEUS to perform the task and limb discrimination needed for successful MTRL continuous control policies. For example, a model can distinguish left from right, not from structural biases as in SMP, but from the relative position of the limb w.r.t. the root node provided in the node features.

While the total number of tasks in the SMP benchmarks is high, they all share one key characteristic. All tasks in a benchmark are built using subsets of the limbs from an archetype (e.g., `Walker++` or `Cheetah++`). To verify that our results hold more broadly, we adapted the `Walkers` benchmark (Wang et al., 2018) and compared AMORPHEUS with SMP and NERVENET on it. This benchmark includes five agents with different morphologies: a Hopper, a HalfCheetah, a FullCheetah, a Walker, and an Ostrich. The results in Figure 4 are consistent[1] with our previous experiments, demonstrating the benefits of AMORPHEUS' fully-connected graph with attentional aggregation.

---

[1]Note that the performance of NERVENET is not directly comparable, as the observational features and the learning algorithm differ from AMORPHEUS and SMP. We do not test NERVENET on SMP benchmarks because the codebases are not compatible and comparing NERVENET and SMP is not the focus of the paper. Even if we implemented NERVENET in the SMP training loop, it is unclear how the critic of NERVENET would perform in a new setting. The original paper considers two options for the critic: one GNN-based and one MLP-based. We use the latter in Figure 4 as the former takes only the root node output labels as an input and is thus most likely to face difficulty in learning multi-hop message-passing. The MLP critic should perform better because training an MLP is easier, though it might be sample-inefficient when the number of tasks is large. For example, in `Cheetah++` an agent would need to learn 12 different critics. Finally, NERVENET learns a separate MLP encoder per task, partially defeating the purpose of using GNN for incompatible environments.

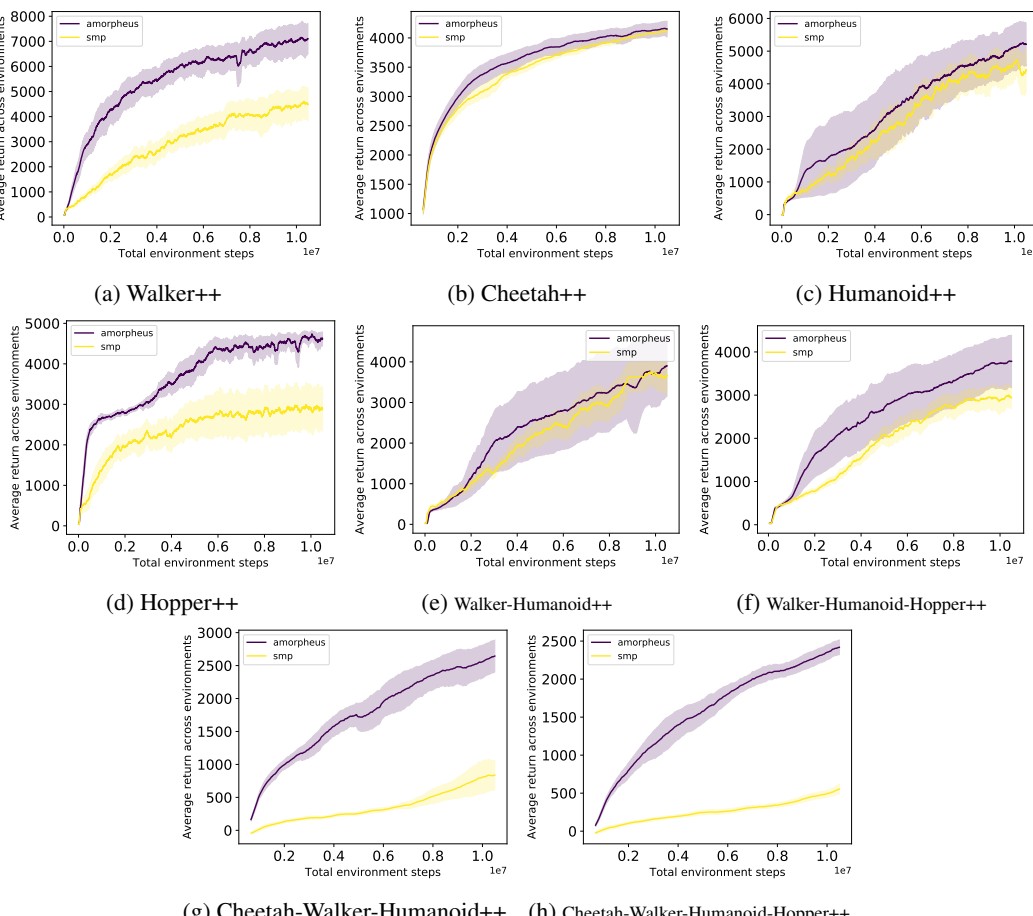

(a) Walker++     (b) Cheetah++     (c) Humanoid++

(d) Hopper++     (e) Walker-Humanoid++     (f) Walker-Humanoid-Hopper++

(g) Cheetah-Walker-Humanoid++    (h) Cheetah-Walker-Humanoid-Hopper++

Figure 3: AMORPHEUS consistently outperforms SMP on MTRL benchmarks from Huang et al. (2020), supporting our hypothesis that no explicit structural information is needed to learn a successful MTRL policy and that facilitated message-passing procedure results in faster learning.

While we focused on MTRL in this work, we also evaluated AMORPHEUS in a zero-shot generalisation setting. Table 3 in Appendix D provides initial results demonstrating AMORPHEUS's potential.

## 4.2 ATTENTION MASK ANALYSIS

GNN-based policies, especially those that use attention, are more interpretable than monolithic MLP policies. We now analyse the attention masks that AMORPHEUS learns. Having an implicit structure that is state dependent is one of the benefits of AMORPHEUS (every node has access to other nodes' annotations, and the aggregation weights depend on the input as shown in Equation 1). By contrast, NERVENET and SMP have a rigid message-passing structure that does not change throughout training or throughout a rollout. Indeed, Figure 5 shows a variety of masks a `Walker++` model exhibits within a `Walker-7` rollout, confirming that AMORPHEUS attends to different parts of the state space based on the input.

Both Wang et al. (2018) and Huang et al. (2020) notice periodic patterns arising in their models. Smilarly, AMORPHEUS demonstrates cycles in attention masks, usually arising for the first layer of the transformer. Figure 6 shows the column-wise sum of the attention masks coordinated with an upper-leg limb of a `Walker-7` agent. Intuitively, the column-wise sum shows how much other nodes are interested in the node corresponding to that column.

Interestingly, attention masks in earlier layers change more slowly within a rollout than those of the downstream layers. Figure 13 in Appendix C.2 demonstrates this phenomenon for three different

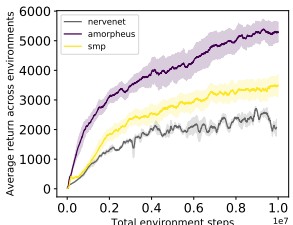 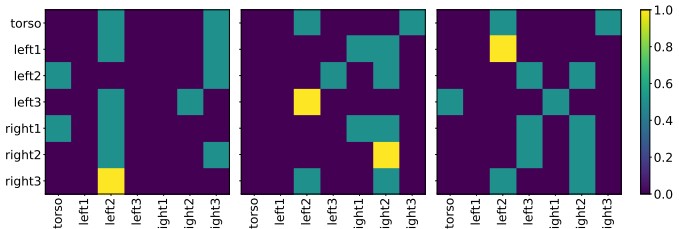

Figure 4: MTRL performance on `Walkers` (Wang et al., 2018).

Figure 5: State-dependent masks of AMORPHEUS (3rd attention layer) within a `Walker-7` rollout.

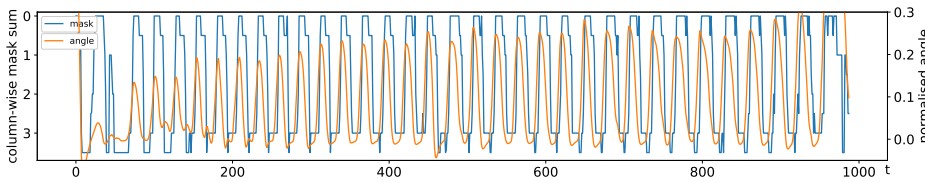

Figure 6: In the first attention layer of a `Walker-7` rollout, nodes attend to an upper leg (column-wise mask sum $\sim 3$) when the leg is closer to the ground (normalized angle $\sim 0$).

`Walker++` models tested on `Walker-7`. This shows that AMORPHEUS might, in principle, learn a rigid structure (as in GNNs) if needed.

Finally, we investigate how attention masks evolve over time. Early in training, the masks are spread across the whole graph. Later on, the mask weights distributions become less uniform. Figures 10, 11 and 12 in Appendix C.1 demonstrate this phenomenon on `Walker-7`.

# 5 RELATED WORK

Most MTRL research considers the compatible case (Rusu et al., 2016; Parisotto et al., 2016; Teh et al., 2017; Vithayathil Varghese & Mahmoud, 2020). MTRL for continuous control is often done from pixels with CNNs solving part of the compatibility issue. DMLab (Beattie et al., 2016) is a popular choice when learning from pixels with a compatible action space shared across the environments (Hessel et al., 2019; Song et al., 2020).

GNNs started to stretch the possibilities of RL allowing MTRL in incompatible environments. Khalil et al. (2017) learn combinatorial optimisation algorithms over graphs. Kurin et al. (2020) learn a branching heuristic of a SAT solver. Applying approximations schemes typically used in RL to these settings is impossible, because they expect input and output to be of fixed size. Another form of (potentially incompatible) RL using message passing are coordination graphs (e.g. DCG, Boehmer et al., 2020), that use the max-plus algorithm (Pearl, 1989) to coordinate action selection between multiple agents. One can apply DCG in single-agent RL using ideas of Tavakoli et al. (2021).

Several methods for incompatible continuous control have also been proposed. Chen et al. (2018) pad the state vector with zeros to have the same dimensionality for robots with different number of joints, and condition the policy on the hardware information of the agent. D'Eramo et al. (2020) demonstrate a positive effect of learning a common network for multiple tasks, learning a specific encoder and a decoder one per task. We expect this method to suffer from sample-inefficiency because it has to learn separate input and output heads per each task. Moreover, Wang et al. (2018) have a similar implementation of their MTRL baseline showing that GNNs have benefits over MLPs for incompatible control. Huang et al. (2020), whose work is the main baseline in this paper, apply a GNN-like approach and study its MTRL and generalisation properties. The method can be used only with trees, its aggregation function is not permutation invariant, and the message-passing schema stays fixed throughout the training procedure. Wang et al. (2018) and Huang et al. (2020) attribute the effectiveness of their methods to the ability of the GNNs to exploit information about agent morphology. In this work, we present evidence against this hypothesis, showing that existing approaches do not exploit morphological information as was previously believed.

Attention mechanisms have also been used in the RL setting. Zambaldi et al. (2018) consider self-attention to deal with an object-oriented state space. They further generalize this to variable action spaces and test generalisation on Starcraft-II mini-games that have a varying number of units and other environmental entities. Duan et al. (2017) apply attention for both temporal dependency and a factorised state space (different objects in the scene) keeping the action space compatible. Parisotto et al. (2020) use transformers as a replacement for a recurrent policy. Loynd et al. (2020) use transformers to add history dependence in a POMDP as well as for factored observations, having a node per game object. The authors do not consider a factored action space, with the policy receiving the aggregated information of the graph after the message passing ends. Baker et al. (2020) use self-attention to account for a factored state-space to attend over objects or other agents in the scene. AMORPHEUS does not use a transformer for recurrency but for the factored state and action spaces, with each non-torso node having an action output. Iqbal & Sha (2019) apply attention to generalise MTRL multi-agent policies over varying environmental objects and Iqbal et al. (2020) extend this to a factored action space by summarising the values of all agents with a mixing network (Rashid et al., 2020). Li et al. (2020) learn embeddings for a multi-agent actor-critic architecture by generating the weights of a graph convolutional network (GCN, Kipf & Welling, 2017) with attention. This allows a different topology in every state, similar to AMORPHEUS, which goes one step further and allows to change the topology in every round of message passing.

Another line of work aims to infer graph topology instead of hardcoding one. Differentiable Graph Module (Kazi et al., 2020) predicts edge probabilities doing a continuous relaxation of k-nearest neighbours to differentiate the output with respect to the edges in the graph. Johnson et al. (2020) learn to augment a given graph with additional edges to improve the performance of a downstream task. Kipf et al. (2018) use variational autoencoders (Kingma & Welling, 2014) using a GNN for reconstruction. Notably, the authors notice that message passing on a fully connected graph might work better than when restricted by skeleton when evaluated on human motion capture data.

## 6 Conclusions and Future Work

In this paper, we investigated the role of explicit morphological information in graph-based continous control. We ablated existing methods SMP and NERVENET, providing evidence against the belief that these methods improve performance by exploiting explicit morphological structure encoded in graph edges. Motivated by our findings, we presented AMORPHEUS, a transformer-based method for MTRL in incompatible environments. AMORPHEUS obviates the need to propagate messages far away in the graph and can attend to different regions of the observations depending on the input and the particular point in training. As a result, AMORPHEUS clearly outperforms existing work in incompatible continuous control. In addition, AMORPHEUS exhibits non-trivial behaviour such as periodic cycles of attention masks coordinated with the gait. The results show that information in the node features alone is enough to learn a successful MTRL policy. We believe our results further push the boundaries of incompatible MTRL and provide valuable insights for further progress.

One possible drawback of AMORPHEUS is its computational complexity. Transformers suffer from quadratic complexity in the number of nodes with a growing body of work addressing this issue (Tay et al., 2020). However, the number of the nodes in continuous control problems is relatively low compared to much longer sequences used in NLP (Devlin et al., 2019). Moreover, Transformers are higly parallelisable, compared to SMP with the data dependency across tree levels (the tree is processed level by level with each level taking the output of the previous level as an input).

We focused on investigating the effect of injecting explicit morphological information into the model. However, there are also opportunities to improve the learning algorithm itself. Potential directions of improvement include averaging gradients instead of performing sequential task updates, or balancing tasks updates with multi-armed bandits or PopArt (Hessel et al., 2019).

### Acknowledgments

VK is a doctoral student at the University of Oxford funded by Samsung R&D Institute UK through the AIMS program. SW has received funding from the European Research Council under the European Union's Horizon 2020 research and innovation programme (grant agreement number 637713). The experiments were made possible by a generous equipment grant from NVIDIA. The authors would like to thank Henry Kenlay and Marc Brockschmidt for useful discussions on GNNs.

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

## A  REPRODUCIBILITY

We initially took the transformer implementation from the Official Pytorch Tutorial (Sequence-to-Sequence Modeling, Pytorch Tutorial) which uses `TransformerEncoderLayer` from Pytorch (Paszke et al., 2017). We modified it for the regression task instead of classification, and removed masking and the positional encoding. Table 1 provides all the hyperparameters needed to replicate our experiments.

Table 1: Hyperparameters of our experiments

| Hyperparameter | Value | Comment |
|---|---|---|
| AMORPHEUS | | |
| – Learning rate | 0.0001 | |
| – Gradient clipping | 0.1 | |
| – Normalisation | LayerNorm | As an argument to `TransformerEncoder` in `torch.nn` |
| – Attention layers | 3 | |
| – Attention heads | 2 | |
| – Attention hidden size | 256 | |
| – Encoder output size | 128 | |
| Training | | |
| – runs | 3 | per benchmark |

AMORPHEUS makes use of gradient clipping and a smaller learning rate. We found, that SMP also performs better with the decreased learning rate (0.0001) as well and we use it throughout the work. Figure 7 demonstrates the effect of a smaller learning rate on `Walker++`. All other SMP hyperparameters are as reported in the original paper with the two-directional message passing.

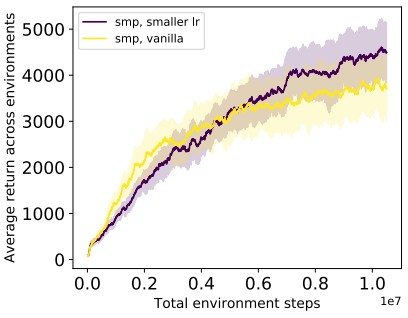

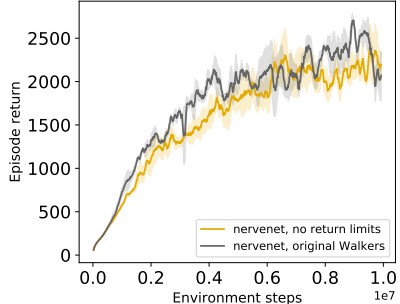

Figure 7: Smaller learning rate make SMP to yield better results on `Walker++`.

Figure 8: Removing the return limit slightly deteriorates the performance of NerveNet on Walkers.

Wang et al. (2018) add an artificial return limit of 3800 for their Walkers environment. We remove this limit and compare the methods without it. For NerveNet, we plot the results with the option best for it. Figure 8 compares the two options.

Table 2: Full list of environments used in this work.

| Environment | Training | Zero-shot testing |
|---|---|---|
| `Walker++` | | |
| | `walker_2_main` | `walker_3_main` |
| | `walker_4_main` | `walker_6_main` |
| | `walker_5_main` | |
| | `walker_7_main` | |
| `humanoid++` | | |
| | `humanoid_2d_7_left_arm` | `humanoid_2d_7_left_leg` |
| | `humanoid_2d_7_lower_arms` | `humanoid_2d_8_right_knee` |
| | `humanoid_2d_7_right_arm` | |
| | `humanoid_2d_7_right_leg` | |
| | `humanoid_2d_8_left_knee` | |
| | `humanoid_2d_9_full` | |
| `Cheetah++` | | |
| | `cheetah_2_back` | `cheetah_3_balanced` |
| | `cheetah_2_front` | `cheetah_5_back` |
| | `cheetah_3_back` | `cheetah_6_front` |
| | `cheetah_3_front` | |
| | `cheetah_4_allback` | |
| | `cheetah_4_allfront` | |
| | `cheetah_4_back` | |
| | `cheetah_4_front` | |
| | `cheetah_5_balanced` | |
| | `cheetah_5_front` | |
| | `cheetah_6_back` | |
| | `cheetah_7_full` | |
| `Cheetah-Walker-` `-Humanoid` | | |
| | All in the column above | All in the column above |
| `Hopper++` | | |
| | `hopper_3` | |
| | `hopper_4` | |
| | `hopper_5` | |
| `Cheetah-Walker-` `-Humanoid-Hopper` | | |
| | All in the column above | All in the column above |
| `Walkers` from Wang et al. (2018) | | |
| | `Ostrich` | |
| | `HalfCheetah` | |
| | `FullCheetah` | |
| | `Hopper` | |
| | `HalfHumanoid` | |

## B    MORPHOLOGY ABLATIONS

Figure 9 shows examples of graph topologies we used in structure ablation experiments.

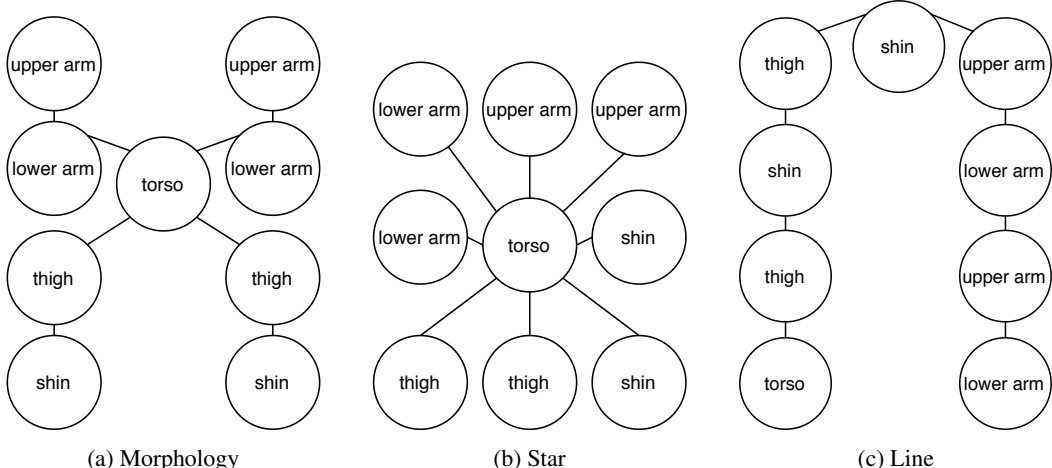

(a) Morphology        (b) Star        (c) Line

Figure 9: Examples of graph topologies used in the structure ablation experiments.

# C  ATTENTION MASK ANALYSIS

## C.1  EVOLUTION OF MASKS THROUGHOUT THE TRAINING PROCESS

Figures 10, 11 and 12 demonstrate the evolution of AMORPHEUS attention masks during training.

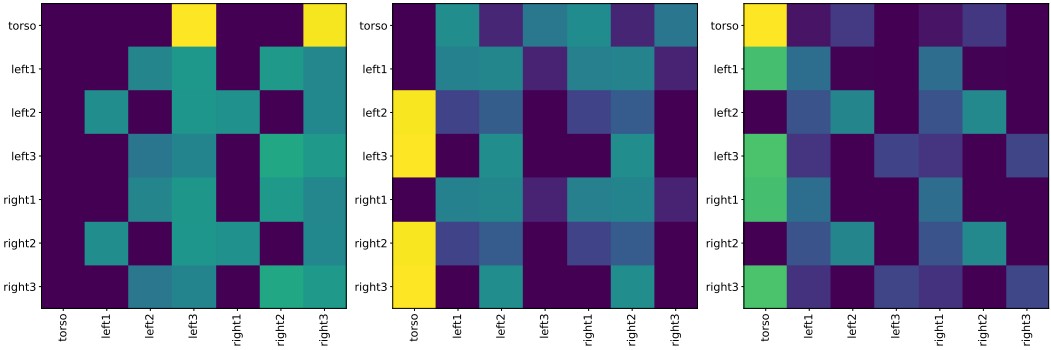

Figure 10: `Walker++` masks for the 3 attention layers on `Walker-7` at the beginning of training.

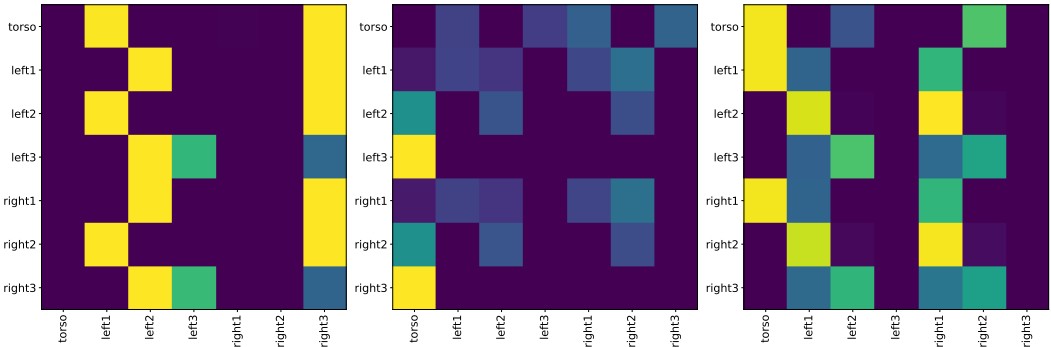

Figure 11: `Walker++` masks for the 3 attention layers on `Walker-7` after 2.5 mil frames.

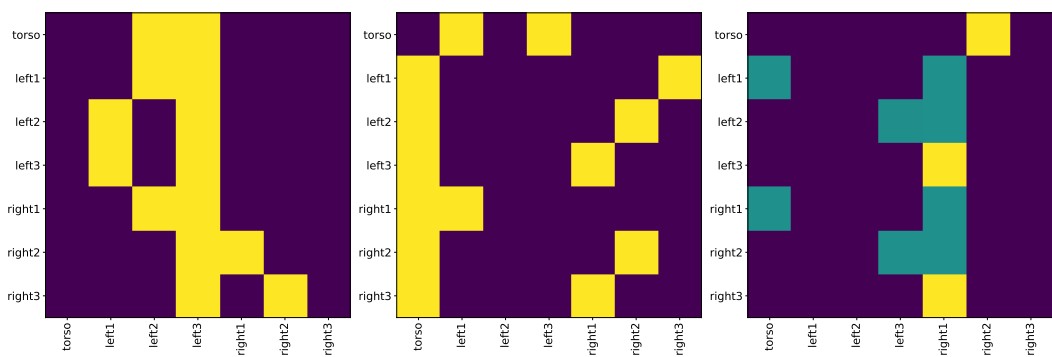

Figure 12: `Walker++` masks for the 3 attention layers on `Walker-7` at the end of training.

### C.2 ATTENTION MASKS CUMULATIVE CHANGE

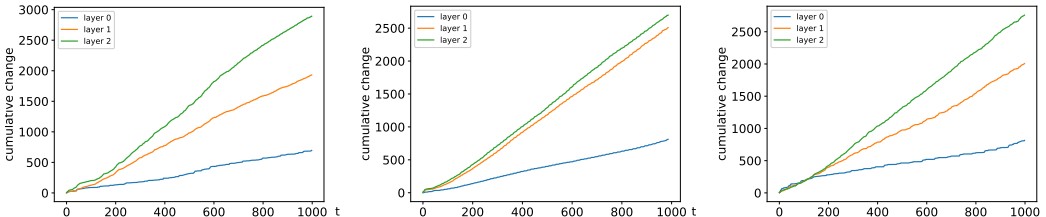

Figure 13: Absolutive cumulative change in the attention masks for three different models on `Walker-7`.

## D GENERALISATION RESULTS

Table 3: Initial results on generalisation. The numbers show the average performance of three seeds evaluated on 100 rollouts and standard error of the mean. While the average values are higher for AMORPHEUS on 5 out of 7 benchmarks, high variance of both methods might be indicative of instabilities in generalisation behaviour due to large differences between the training and testing tasks.

|  | AMORPHEUS | SMP |
|---|---|---|
| `walker-3-main` | **666.24** (133.66) | 175.65 (157.38) |
| `walker-6-main` | **1171.35** (832.91) | 729.26 (135.60) |
| `humanoid-2d-7-left-leg` | **2821.22** (1340.29) | 2158.29 (785.33) |
| `humanoid-2d-8-right-knee` | **2717.21** (624.80 ) | 327.93 (125.75) |
| `cheetah-3-balanced` | **474.82** (74.05) | 156.16 (33.00) |
| `cheetah-5-back` | 3417.72 (306.84) | **3820.77** (301.95) |
| `cheetah-6-front` | 5081.71 (391.08) | **6019.07** (506.55) |

## E RESIDUAL CONNECTION ABLATION

We use the residual connection in AMORPHEUS as a safety mechanim to prevent nodes from forgetting their own observations. To check that AMORPHEUS's improvements do not come from the residual connection alone, we performed the ablation. As one can see on Figure 14, we cannot attribute the success of our method to this improvement alone. High variance on `Humanoid++` is related to the fact that one seed started to improve much later, and the average performance suffered as the result.

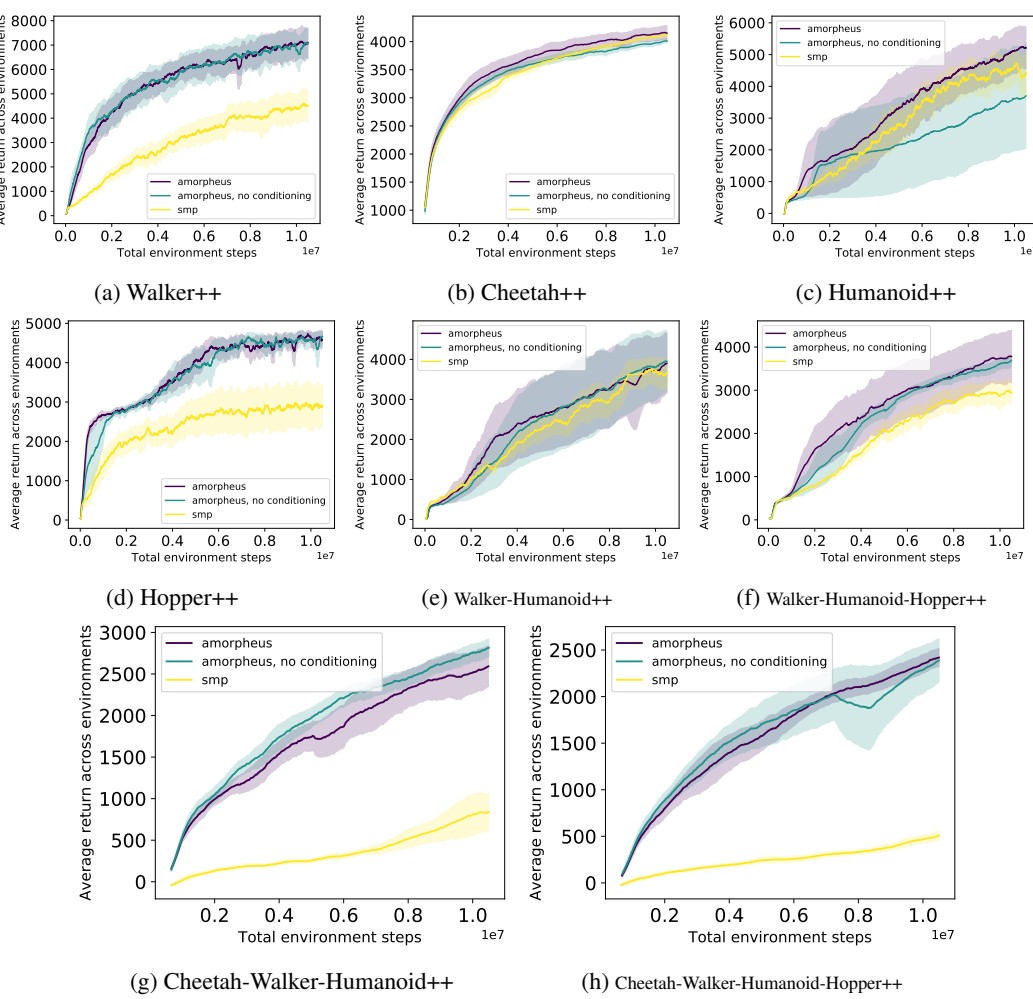

Figure 14: Residual connection ablation experiment.

