# OpenReview forum: "My Body is a Cage: the Role of Morphology in Graph-Based Incompatible Control"
_ICLR.cc/2021/Conference — ICLR 2021 Poster_

### Official Review · AnonReviewer4 · 2020-10-26
**Clear contributions to expanding the capabilities of RL agents through GNNs, and transformers in particular.**

**Rating:** 7
**Confidence:** 4

**Review:**

This work considers continuous control environments in which each agent limb (actuator) is associated with one action and a set of observation factors. As in prior work, the proposed policy class is modular, where each module is mapped to one limb (or the root), and the modules share information through some GNN message-passing schedule. The experimental results indicate that fully connected, transformer-style message passing is more effective in this setting than message passing restricted to directly connected pairs of limbs.

Pros
- Good framing of the problem and choice of experiments.
- Insightful discussion of related work.

Cons
- There is no discussion of the additional computational requirements of transformers over SMP.
- The results would be stronger if hyperparameters had been systematically tuned.

Questions
- Why were no results provided for the Walker-Hopper or Walker-Hopper-Humanoid combinations tested by Huang et al?

Suggestions
- The paper mentions in passing that this work involves agents “with each non-torso node having an action output”. This limitation probably deserves to be highlighted more prominently.

- As the paper explains, “transformers can be seen as GNNs operating on fully connected graphs”. In other places, the paper contrasts transformers with GNN-based methods (“substantially outperforms GNN-based methods”), as if transformers were not GNNs. To avoid confusing readers, it would help to explain that GNNs are a broad class that includes both transformers and SMP, which differ in their message passing schedules, etc.

- It would help to have a more detailed description of the limb torques and observation factors, so that readers don’t have to look in Huang et al (section 4) for these details.

- The following phrases are important but unclear for readers who are not very familiar with GNNs or transformers:  “Having an implicit structure that is state dependent is one of the benefits of AMORPHEUS.”  and  “the implicit state-dependent message-passing schema learnt by AMORPHEUS can be better”

- The paper says “We use entity to denote both vertices and edges.” But the term “entity” appears nowhere else in the paper.

---

> ### Author Response · Authors · 2020-11-13
> **Response to AnonReviewer4**
>
> Thanks for your constructive feedback! We updated our paper with clarifications to address the specific comments below.
>
>
> * There is no discussion of the additional computational requirements of transformers over SMP.
>     * We added a paragraph discussing this to Section 6.
> * The results would be stronger if hyperparameters had been systematically tuned.
>     * We fully agree, however, this would require prohibitive computational resources (MTRL experiments are computationally demanding, e.g. Cheetah-Walker-Humanoid-Hopper benchmark has 25 environments in it), and we believe that pushing the return curve even higher would not change our conclusions.
> * Why were no results provided for the Walker-Hopper or Walker-Hopper-Humanoid combinations tested by Huang et al?
>     * Huang et al do not include Cheetah++ in their cross-benchmark experiments because it makes their training unstable (they claim this is caused by different integrator in MuJoCo). As we show, Amorpheus does not suffer from the same behaviour and considerably outperforms SMP on the full benchmark suite (Cheetah-Walker-Humanoid-Hopper). For completeness, we’ve started the experiments on Walker-Hopper and Walker-Hopper-Humanoid and will update the paper with the plots when they are ready.
> * The paper mentions in passing that this work involves agents “with each non-torso node having an action output”. This limitation probably deserves to be highlighted more prominently.
>     * This is not really a limitation since the network still has an output for the torso, it’s just the environment does not need it (and we simply ignore it just like in the SMP paper)
> * As the paper explains, “transformers can be seen as GNNs operating on fully connected graphs”. In other places, the paper contrasts transformers with GNN-based methods (“substantially outperforms GNN-based methods”), as if transformers were not GNNs. To avoid confusing readers, it would help to explain that GNNs are a broad class that includes both transformers and SMP, which differ in their message passing schedules, etc.
>     * We clarified this in the paper referring to SMP/NerveNet as GNNs that use morphological information to define the message-passing scheme.
> * It would help to have a more detailed description of the limb torques and observation factors, so that readers don’t have to look in Huang et al (section 4) for these details.
>     * We added a full description of the state space to Section 4.
> * The following phrases are important but unclear for readers who are not very familiar with GNNs or transformers: “Having an implicit structure that is state dependent is one of the benefits of AMORPHEUS.” and “the implicit state-dependent message-passing     schema learnt by AMORPHEUS can be better”
>     * Thank you for pointing this out. We clarified the first statement it in the paper relating to Equation 1 in Section 2.3. We reformulated the second statement.
> * The paper says “We use entity to denote both vertices and edges.” But the term “entity” appears nowhere else in the paper.
>     * We removed this sentence.

---

> > ### Comment · AnonReviewer4 · 2020-11-25
> > **Response**
> >
> > Thank you for making these improvements to a great paper!

---

### Official Review · AnonReviewer1 · 2020-10-27
**Strong performance against baselines; architecture/experiment details less clear**

**Rating:** 7
**Confidence:** 4

**Review:**

The paper focuses on the problem of multi-task control with a shared policy in the continuous action setting. Unlike current assumption of compatible state-action spaces, the proposed architecture is transferable across different morphologies. The paper includes ablation experiments that clearly show that current works that use the body morphology structure to constrain the graph structure of graph neural network based approaches do not actually improve the performance. The paper instead forgoes trying to input the body structure and uses a transformer based architecture that is capable of learning the appropriate (even dynamic) graph structure actually useful for control.

I enjoyed the relatively simple experiments showing how the specific graph struture based on body morphology wasn't important at all. Although would be useful to know how many runs were performed given the noisy nature of RL.
Similarly the cyclical structure noticed in Fig. 6 definitely points towards the powerful nature of transformer architectures at learning this relations. The strong performances (none of which seem to have converged yet) compared to baselines speak for themselves. Although, again not clear about the number of seeds the experiments were repeated.

The architecture description is somewhat unclear. Both actor and critic seem to have three parts to their architecture. But for a critic you need to output value information which might be scalar unlike decoder MLPs for independent node action. If there is some sort of aggregation going on, it needs to be clarified as to specifically how.
Although Fig. 5 shows changing attention patterns, it doesn't warrant the confirmation that the proposed architecture benefits from "state-dependent message passing of transformers" which itself consists of two things. One can do state-dependent message passing in such architectures without the transformers (see DICG [1] for an example with attention and graph convolutions). Second, there could be a static graph structure which is better than the dynamic masks: the paper didn't actually perform the experiments to rule that out. Maybe the obvious morphology is the wrong graph structure but there is something else which would work better.
Again, the paper's claim is probably true, but the causal language is not justified from Fig 5.

[1] https://arxiv.org/abs/2006.11438

Edit: Updated score to reflect the changes from the revision.

---

> ### Author Response · Authors · 2020-11-13
> **Response to AnonReviewer1**
>
> Thanks for your constructive feedback! We updated our paper to address the specific comments below.
>
> * The strong performances (none of which seem to have converged yet) compared to baselines speak for themselves. Although, again not clear about the number of seeds the experiments were repeated.
>     * We added this to the revision. Similarly to NerveNet (3 seeds) and SMP (4 seeds), we use 3 seeds for every experiment. It is difficult to use more seeds in MTRL as the experiments are much more computationally demanding compared to single-task RL. For example, Cheetah-Walker-Humanoid-Hopper experiments imply training on 25 environments.
> * The architecture description is somewhat unclear. Both actor and critic seem to have three parts to their architecture. But for a critic you need to output value information which might be scalar unlike decoder MLPs for independent node action. If there is some sort of aggregation going on, it needs to be clarified as to specifically how.
>     * We clarified this in the latest revision. As we mention in the paper, we took the training loop from the SMP codebase and replaced the models only. We use both actor and a critic the same way SMP uses them: actor and critic are two independent networks (for us it is transformers). Same as in SMP paper, we output a scalar per node which is used as an action for the policy network, and as a value for the critic. There is no aggregation at the critic level, and the value loss is computed per each node independently (Though using the same scalar reward from the environment to compute the targets).
> * Although Fig. 5 shows changing attention patterns, it doesn't warrant the confirmation that the proposed architecture benefits from "state-dependent message passing of transformers" which itself consists of two things. One can do state-dependent message passing in such architectures without the transformers (see DICG [1] for an example with attention and graph convolutions). Second, there could be a static graph structure which is better than the dynamic masks: the paper didn't actually perform the experiments to rule that out. Maybe the obvious morphology is the wrong graph structure but there is something else which would work better. Again, the paper's claim is probably true, but the causal language is not justified from Fig 5.
>     * Thanks for the DICG reference! We agree with this point and reformulated this claim in the paper. From our experiments, we cannot exclude the possibility that there is no optimal fixed static graph. However, even if this is the case, transformers could in principle learn that as well.

---

### Official Review · AnonReviewer2 · 2020-10-28
**Interesting paper, needs a bit more analysis**

**Rating:** 7
**Confidence:** 4

**Review:**

This paper proposes that recent methods that used graphical neural networks to help solve the multitask reinforcement learning problem and assume that there's an advantage from being able to encode the agent's morphology using a graphical neural network do not provide additional generalization and benefits for learning. Instead, they claim that the benefits from being able to encode this morphology are counteracted by the difficulty in having to train the graphical neural network using the message passing system. This paper instead proposes to use Transformers as a simpler mechanism to be able to train and discover the helpful morphological distinctions between agents in order to better solve multitask reinforcement learning problems.

The motivation that graphical neural networks are bogged down by their message passing framework is not necessarily a motivation for using transformers. There needs to be a separate motivation for why you want to use transformers and why they should perform better than GNNs or normal networks.

The author claims that the SMP paper does not work better due to the morphology encoding and then they point out that it instead works because of the encoding of the subtrees and some specific detail related to message passing. This could be correct the explanation but the paper so far hasn't gone into enough detail for the reader to understand the importance of this message passing and how it works and how it is not improving training for GNNs.

Figure one does provide some information related to how the transformer is used with respect to some morphology it would be far more helpful if this figure method was described well enough so that anyone can understand how to apply this to a different morphology. One of the challenges with reading and understanding this paper is the lack of information on how graphical neural networks are used and designed to understand later comments in the paper.

There needs to be an ablation with respect to the residual connections added to the Transformer based network to make sure the improvement for amorphous is not working well just because of these residual connections.

While I do agree that training a graphical neural network to be able to produce a quality policy for a number of control tasks from the opening item environment is difficult the author of the paper might be missing at least one of the key points from the previous work in that you can learn a stronger modularization of policy. And that a goal of the SMP work was to understand how more modular policies or policies with modular components could be learned.

It is stated in the paper that amorphous does better for state of the art incompatible continuous control? What is meant by incompatible continuous control? This term has not been defined anywhere in the paper and without this definition, it's difficult to understand the contribution this paper is making.

----- Post Discussion ----
I have updated my rating for the paper after the authors have provided additional discussion and experiments to address my concerns.

---

> ### Author Response · Authors · 2020-11-13
> **Response to AnonReviewer2**
>
> Thanks for your constructive feedback! We updated our paper to address the specific comments below.
>
> * The motivation that graphical neural networks are bogged down by their message passing framework is not necessarily a motivation for using transformers. There needs to be a separate motivation for why you want to use transformers and why they should perform better than GNNs or normal networks.
>     * We believe we provide a strong motivation for using transformers in our work: GNNs are modular and allow encode additional information in the graph structure. However, as we show, physical morphology information does not affect the performance of NerveNet/SMP, hence, we can use simpler (from the message passing perspective) models which preserve modularity (i.e. Transformers, as opposed to normal networks can be used across incompatible environments) , but have a computation graph that is not restricted by the input graph topology.
> * The author claims that the SMP paper does not work better due to the morphology encoding and then they point out that it instead works because of the encoding of the subtrees and some specific detail related to message passing. This could be the correct explanation but the paper so far hasn't gone into enough detail for the reader to understand the importance of this message passing and how it works and how it is not improving training for GNNs.
>     * We moved the factual information to the background section (detail on message passing) and made it clear that we do not provide an explanation of why SMP really works.
> * Figure one does provide some information related to how the transformer is used with respect to some morphology it would be far more helpful if this figure method was described well enough so that anyone can understand how to apply this to a different morphology. One of the challenges with reading and understanding this paper is the lack of information on how graphical neural networks are used and designed to understand later comments in the paper.
>     * We updated Section 4 to clarify how we use the transformers are used, and what the state space includes. As we stress in the paper, we use exactly the same state representation as SMP as well as the way policy/critic are used (as explained insection 2.2).
> * There needs to be an ablation with respect to the residual connections added to the Transformer based network to make sure the improvement for amorphous is not working well just because of these residual connections.
>     * Thank you for pointing this out! We originally added the residual connections as a safety mechanism for the nodes to prevent forgetting their own observations. However, we ran additional ablations that show that skip connections do not change Amorpheus' performance, and we therefore will remove them from the newest version of the paper for simplicity. The affected plots will be updated as soon as all experiments have finished.
> * While I do agree that training a graphical neural network to be able to produce a quality policy for a number of control tasks from the opening item environment is difficult the author of the paper might be missing at least one of the key points from the previous work in that you can learn a stronger modularization of policy. And that a goal of the SMP work was to understand how more modular policies or policies with modular components could be learned.
>     * As AnonReviewer4 points out, Amorpheus also learns a modular policy similar to NerveNet and SMP. The main goal of our work was to show that one of the assumptions of SMP/NerveNet does not hold, and removing this assumption and using transformers yields higher returns in the MTRL setting. In fact, Amorpheus is even more modular than SMP, because the latter relies on an assumption, that during testing, the agent will have fewer or the same number of children for each of the nodes in the input tree.
> * It is stated in the paper that amorphous does better for state of the art incompatible continuous control? What is meant by incompatible continuous control? This term has not been defined anywhere in the paper and without this definition, it's difficult to understand the contribution this paper is making.
>     * We introduce incompatible control in the introduction and define it formally at the end of Section 2.2: Incompatible MTRL for continuous control implies learning a common policy for a set of agents with different number of limbs and connectivity of those limbs, i.e. morphology. To be more precise, a set of incompatible continuous control environments is a set of MDPs described in Section 2.1.

---

> > ### Comment · AnonReviewer2 · 2020-11-21
> > **Clarifications appreciated**
> >
> > Thank you for updating the text to improve a few rough edges.
> >
> > "ablation with respect to the residual connections"
> > Good to hear. Looking forward to the plots that show this and indicate you can use an even simpler model.
> >
> > "stronger modularization of policy"
> > This point is important. It would help to discuss this more in the paper.
> >
> > Looking over the paper again. In Figure 3 there is little difference between the methods for the cheetah and humanoid++ environments. Is there a reason for this? One might expect that part of the reason the included morphology prior to SMP does not help is that the morphology for some problems is not complex. As the morphology gets more complex the use of this prior may be more helpful.

---

> > > ### Author Response · Authors · 2020-11-22
> > > **Ablations uploaded**
> > >
> > > - "ablation with respect to the residual connections" Good to hear. Looking forward to the plots that show this and indicate you can use an even simpler model.
> > >     - Please, find the ablation in the Appendix E. From the plots, one can see that residual connection alone cannot be the underlying factor of Amorpheus' success. However, given the results on Humanoid++, we decided to keep the residual connection in place. Increased variance of the run on the plot for Humanoid++ is related to the fact that one of the seeds started to learn much later compared to the others. Ablation for Cheetah-Walker-Hopper-Humanoid is still running (it is the most resource hungry one), and we will update the plot in the final revision. We believe that this does not change our conclusions in any way.
> > >
> > > - "stronger modularization of policy" This point is important. It would help to discuss this more in the paper.
> > >     - We added the following lines to Section 4: “Similarly to NerveNet and SMP, Amorpheus is modular and can be used in incompatible environments, including those not seen in training.
> > > In contrast to SMP which is constrained by the maximum number of children per node defined at the model initialisation in training, Amorpheus can be applied to any other morphology with no constraints on the physical connectivity.”
> > >
> > > - Looking over the paper again. In Figure 3 there is little difference between the methods for the cheetah and humanoid++ environments. Is there a reason for this? One might expect that part of the reason the included morphology prior to SMP does not help is that the morphology for some problems is not complex. As the morphology gets more complex the use of this prior may be more helpful.
> > >     - This is an interesting point, but it is not clear whether this is due to the morphology’s complexity. For example, Humanoid should be a complex body-type, but Figure 1b clearly demonstrates that the morphology is not helping SMP. While SMP shows the same effect for Walker in Figure 1a, Amorpheus significantly improves the performance for this body type. This indicates that it might make more sense to think of complexity as the variety of gaits different simultaneously trained morphologies require. This may explain why Hopper++, one of the simplest possible morphologies in a traditional sense, improves significantly with Amorpheus. This would also explain why the advantage of Amorpheus is clearer the more different body-types are included during training, i.e. in Cheetah-Walker-Humanoid(-Hopper)++.

---

> > > > ### Comment · AnonReviewer2 · 2020-11-25
> > > > **Feedback**
> > > >
> > > > The experiments in Appendix E are very helpful in clearing up my concerns.
> > > >
> > > > "As the morphology gets more complex the use of this prior may be more helpful."
> > > > - These comments are appreciated. This complexity is not related to the claims in the paper.
> > > >
> > > > I have increased my rating for the paper.

---

### Official Review · AnonReviewer3 · 2020-10-31
**Broken body in healthy mind - an interesting apporach with potential**

**Rating:** 7
**Confidence:** 2

**Review:**

The manuscript studies the usefulness of Graph Neural Networks (GNN) in incomplete environments for Multitask Reinforcement Learning (MTRL). First, authors explore to what extend morphology information improves performance in GNN. By use of the Shared Modular Policies and the NerveNet methods, authors find that restricting morphology information does not improve performances. Based on this finding, authors apply GNN to fully connected graphs with memory/attention, i.e. the use transformers. Simulation results in different environments show that the proposed approach outperforms conventional methods.

The paper is well written, and methods and analysis approach used are clear. The selected approach and the authors findings are meaningful in a neuroscientific context, as the transformers approach better resembles the function of a human brain. The contribution is original and relevant for the community.

The introduction does not clearly explain why the assumption that restricting the model and encoding morphological information may be beneficial. A weakness of the manuscript is the assessment of the results. No statistical tests were calculated to support the authors claims, nor is it clear how the performance of the different methods was compared. For example, the comparison f the curves depicted in Figure 3. It is also not clear to the reviewer what specifically the highlighted areas in the figures represents? Confidence intervals or standard deviation? Moreover, a few statements are vague. For example, what is a “good MTRL policy” (page 6).

Overall an interesting approach that is expected to improve performance in MTRL and needs further exploring.

---

> ### Author Response · Authors · 2020-11-13
> **response to AnonReviewer3**
>
> Thanks for your constructive feedback! We updated our paper to address the specific comments below.
> * The introduction does not clearly explain why the assumption that restricting the model and encoding morphological information may be beneficial.
>     * This is an assumption of prior literature, which we empirically disprove, as seen in Figure 1. We emphasized this more clearly in the introduction. At first glance, it is natural to use the physical morphology to construct the state graph: nearby joints are close in the graph; communication reflects the kinematic chain and additional information tells the GNN the morphology. However, as we show, these things are not advantageous.
>
> * No statistical tests were calculated to support the authors’ claims, nor is it clear how the performance of the different methods was compared. For example, the comparison of the curves depicted in Figure 3.
>     * We follow a standard protocol adopted in most of the Deep RL works: compare the training curves which plot the agent mean and standard error against the number of environment steps the agent experiences. Due to computational constraints and the large number of environments per random seed, we only report the mean and standard error of 3 seeds. However, in experiments like Cheetah-Walker-Humanoid-Hopper++ (Figure 3f), Amorpheus outperforms SMP by 8+ standard deviations, which is clearly statistically significant."
>
> * It is also not clear to the reviewer what specifically the highlighted areas in the figures represents? Confidence intervals or standard deviation?
>     * The shades on the plots are standard error of the mean, which is standard practice in Deep RL. We included this in the paper now.
>
> * Moreover, a few statements are vague. For example, what is a “good MTRL policy” (page 6).
>     * By “good MTRL policy” we mean a policy achieving high average return across a set of environments as defined in the second paragraph of Section 2.1 ($\frac{1}{N}\sum_{i=1}^N{J_i}$). We rephrased the statement in the revision.

---

> > ### Comment · AnonReviewer3 · 2020-11-24
> > **Thanks for clarifications**
> >
> > Thank you for adding details. It is clearer now.  Still a good paper.
> >
> > Related to statistical significance the reviewer would like to note that computing a statistical significance test and comparing the results to random performance is needed to ensure that the results are significanlty better and useful.

---

### Decision · Program_Chairs · 2021-01-07
**Final Decision**

**Decision:**

Accept (Poster)

**Comment:**

The paper shows that using graph neural networks to address multi-task control problems with incompatible environments does not provide benefits to the learning process. The authors instead propose to use Transformers as a simpler mechanism to be able to train and discover the helpful morphological distinctions between agents in order to better solve multitask reinforcement learning problems.

The paper is well written and the analysis of the literature has been appreciated.  The contribution is original and relevant to the community.

All the reviewers agree that this paper deserves acceptance. We invite the authors to modify the paper by following the suggestions provided by the reviewers. In particular:
- improve the analysis of the empirical results
- update the plots
- add the suggested references